

# Changes in the North Atlantic Oscillation over the 20th Century

Stephen Outten[1] and Richard Davy[1]

[1]Nansen Environmental and Remote Sensing Center, Bjerknes Centre for Climate Research, Bergen, NORWAY

**Correspondence:** Stephen Outten (stephen.outten@nersc.no)

**Abstract.** The North Atlantic Oscillation explains a large fraction of the climate variability across the North Atlantic from the eastern seaboard of North America across the whole of Europe. Many studies have linked the North Atlantic Oscillation to climate extremes in this region, especially in winter, which has motivated considerable study of this pattern of variability. However, one overlooked feature of how the North Atlantic Oscillation has changed over time is the explained variance of the pattern. Here we show that there has been a considerable increase in the percentage variance explained by the NAO over the 20th century from 32% in 1930 to 53% by the end of the 20th century. Whether this change is due to natural variability, a forced response to climate change, or some combination remains unclear. However, we found no evidence for a forced response from an ensemble of 50 CMIP6 models. These models did all show substantial internal variability in the strength of the North Atlantic Oscillation, but it was biased towards being too high compared to the reanalysis and with too little variation over time. Since there is a direct connection between the North Atlantic Oscillation and climate extremes over the region, this has direct consequences for both the long term projection and near term prediction of changes to climate extremes in the region.

## 1 Introduction

The North Atlantic Oscillation (NAO) is a pattern of variability in the sea level pressure over the North Atlantic associated with the subpolar low and subtropical high. The NAO is associated with large scale changes in the position and intensity of both the storm track and the jet stream over the North Atlantic, and thus plays a direct role in shaping atmospheric heat and moisture transport across the basin (Fasullo et al., 2020). It has also been shown that the NAO has a large impact on the Atlantic Meridional Overturning Circulation, and thus oceanic heat transport, and that this is largest on timescales of 20-30 years leading to changes in northern hemisphere temperature of several tenths of a degree (Delworth and Zeng, 2016). The NAO has a positive and negative phase, and exhibits considerable interannual variation between the two phases. The positive phase of the NAO indicates deeper than normal low pressure in the subpolar region and higher than normal high pressure over the subtropics. It is often associated with decreased temperature and precipitation anomalies over Southern Europe, and increased precipitation anomalies over Northern Europe. The effects of the NAO are basin wide and the positive phase is also associated with positive anomalies in temperature over the Eastern United States. The opposite pattern and effects are seen during periods when the NAO is in its negative phase (Weisheimer et al., 2017).

It has long been established that the NAO dominates climate variability across a large part of the Northern hemisphere from the Eastern seaboard of North America across Europe, to central Russia; and from the Arctic in the North to the subtropical





Atlantic (Hurrell et al., 2003). The NAO is an especially important component of the winter variability and has been linked to the frequency and intensity of climate extremes over Europe (Haylock and Goodess, 2004; Scaife et al., 2008; Fan et al., 2016). It is therefore essential to understand the scale of natural variability of the NAO, how the NAO responds to changes in

external forcing, and whether these are captured in current climate models. If current climate models fail to capture either the natural variability or the response-to-forcing of the NAO this could lead to a radically different projection of changes to climate extremes over Europe on the decadal to century timescales.

An index for the NAO is often identified in one of two ways. The first approach is to calculate the normalized difference in surface pressure between the subtropical high (Azores High) and subpolar low (Icelandic Low) over the North Atlantic

sector. The second approach is to perform an Empirical Orthogonal Function (EOF) analysis on sea level pressure over the North Atlantic region. EOF analysis separates the variability of the sea level pressure into orthogonal modes, with the first mode containing the largest proportion of the variability, and each subsequent mode containing progressively less. When EOF analysis is used to calculate the NAO, the first mode indicates the NAO index, while the second and third modes usually provide the North Atlantic Ridge and Scandinavian Blocking patterns (Cassou et al., 2004).

An EOF produces two outputs, an eigenvalue and an eigenvector for each mode of variability. The eigenvector or Principal Component (PC) as it is called is a time series showing the variation of the mode in time, while the eigenvalue is a single value that quantifies how much of the variation in the original field is explained by the particular mode. Usually, the eigenvector is regressed back onto the original field to produce a map showing the pattern of variation associated with the mode, while the eigenvalue is weighted with all the eigenvalues to convert it to a percentage of the total variance explained by the given mode.

Most studies using an EOF to examine the NAO focus on the eigenvector, to highlight the pattern of variability, to examine the PC for signs of natural variability, or to study the changes in phase of the NAO as these relate directly to downstream weather changes. However, changes to the eigenvalue of each mode of variability have been largely overlooked in the literature. Changes to the eigenvalue tell us how the relative dominance of each mode has changed over time, either due to changes in the amplitude of the variability along the axis of a given mode or due to changes in the magnitude of the variability of other modes. A key

question we wish to address is whether recent changes to the relative dominance of each of the modes of variability in the North Atlantic is due to changes in external forcing or natural variability. Given the strong influence of the NAO and other atmospheric modes of variability on regional climate in Europe, it is crucial to understand the role of natural variability and any forced response in order to make reliable predictions and projections of future changes to the NAO and hence climate extremes in these regions. In this study we focus on the eigenvalue and investigate how the percentage variability explained by

each mode has changed over time, and how well the latest generation of climate models in CMIP6 capture this change.

## 2   Data Sources

In this study we use monthly sea level pressure data from the European Centre for Medium-Range Weather Forecasts (ECMWF) 20th Century reanalysis (ERA-20C; Poli et al., 2016) and NOAA-CIRES 20th Century Reanalysis V2 (NOAA-20CR; Compo et al., 2011), for the period of 1900 to 2010. The NOAA-CIRES 20th Century Reanalysis (V2) data provided by the NOAA



PSL, Boulder, Colorado, USA, from their website at https://psl.noaa.gov. The NOAA-20CR and ERA-20C reanalyses have horizontal resolutions of 2°and 1°respectively for the atmospheric fields.

The climate model data analysed in this study comes from the Coupled Model Inter-comparison Project Phase 6 (CMIP; Eyring et al., 2016), made publicly available through the Earth System Grid Foundation web portal https://esgf-data.dkrz.de/. We use monthly averages of sea level pressure and precipitation flux from the historical simulations. The climate models

analysed vary in horizontal resolution from 0.5°in the CNRM-CM6-1-HR model to approximately 2.5°in the MCM-UA-1-0 model. Since the Common Basis Function method used in this study requires the climate model data to be on the same grid as the reanalysis used for the basis, the climate models were all interpolated to the 1°grid of the ERA-20C reanalysis using bilinear interpolation. The complete list of CMIP6 models examined in this study is given in Table 1.

| Model Name | Institute | Institute Abbreviation |
| --- | --- | --- |
| ACCESS-CM2 | Commonwealth Scientific and Industrial Research Organisation, Australian Research Council Centre of Excellence for Climate System Science | CSIRO-ARCCSS |
| ACCESS-ESM1-5 | Commonwealth Scientific and Industrial Research Organisation | CSIRO |
| AWI-CM-1-1-MR | Alfred Wegener Institute, Helmholtz Centre for Polar and Marine Research | AWI |
| AWI-ESM-1-1-LR | Alfred Wegener Institute, Helmholtz Centre for Polar and Marine Research | AWI |
| BCC-CSM2-MR | Beijing Climate Center | BCC |
| BCC-ESM1 | Beijing Climate Center | BCC |
| CAMS-CSM1-0 | Chinese Academy of Meteorological Sciences | CAMS |
| CAS-ESM2-0 | Chinese Academy of Sciences | CAS |
| CESM2-FV2 | National Center for Atmospheric Research | NCAR |
| CESM2-WACCM-FV2 | National Center for Atmospheric Research | NCAR |
| CESM2-WACCM | National Center for Atmospheric Research | NCAR |
| CESM2 | National Center for Atmospheric Research | NCAR |
| CMCC-CM2-SR5 | Fondazione Centro Euro-Mediterraneo sui Cambiamenti Climatici | CMCC |
| CMCC-ESM2 | Fondazione Centro Euro-Mediterraneo sui Cambiamenti Climatici | CMCC |
| CNRM-CM6-1-HR | Centre National de Recherches Meteorologiques, Centre Europeen de Recherche et de Formation Avancee en Calcul Scientifique | CNRM-CERFACS |
| CNRM-CM6-1 | Centre National de Recherches Meteorologiques, Centre Europeen de Recherche et de Formation Avancee en Calcul Scientifique | CNRM-CERFACS |
| CNRM-ESM2-1 | Centre National de Recherches Meteorologiques, Centre Europeen de Recherche et de Formation Avancee en Calcul Scientifique | CNRM-CERFACS |
| CanESM5-CanOE | Canadian Centre for Climate Modelling and Analysis, Environment and Climate Change Canada | CCCma CCCma |
| CanESM5 | Canadian Centre for Climate Modelling and Analysis, Environment and Climate Change Canada | CCCma CCCma |
| E3SM-1-0 | E3SM Consortium | E3SM-Project |
| E3SM-1-1 | E3SM Consortium | E3SM-Project |
| EC-Earth3-Veg | EC-Earth Consortium | EC-Earth-Consortium |
| FGOALS-f3-L | Chinese Academy of Sciences | CAS |
| FGOALS-g3 | Chinese Academy of Sciences | CAS |





| FIO-ESM-2-0 | First Institute of Oceanography | FIO-QLNM |
| GFDL-CM4 | National Oceanic and Atmospheric Administration | NOAA-GFDL |
| GFDL-ESM4 | National Oceanic and Atmospheric Administration | NOAA-GFDL |
| GISS-E2-1-G-CC | Goddard Institute for Space Studies | NASA-GISS |
| GISS-E2-1-G | Goddard Institute for Space Studies | NASA-GISS |
| GISS-E2-1-H | Goddard Institute for Space Studies | NASA-GISS |
| HadGEM3-GC31-LL | Met Office Hadley Centre | MOHC |
| HadGEM3-GC31-MM | Met Office Hadley Centre | MOHC |
| INM-CM4-8 | Institute for Numerical Mathematics | INM |
| INM-CM5-0 | Institute for Numerical Mathematics | INM |
| IPSL-CM6A-LR | Institut Pierre Simon Laplace | IPSL |
| KACE-1-0-G | National Institute of Meteorological Sciences/Korea Meteorological Administration | NIMS-KMA |
| MCM-UA-1-0 | Department of Geosciences, University of Arizona | UA |
| MIROC-ES2L | Japan Agency for Marine-Earth Science and Technology | MIROC |
| MIROC6 | Japan Agency for Marine-Earth Science and Technology | MIROC |
| MPI-ESM-1-2-HAM | HAMMOZ Consortium | HAMMOZ-Consortium |
| MPI-ESM1-2-HR | Max Planck Institute for Meteorology | MPI-M |
| MPI-ESM1-2-LR | Max Planck Institute for Meteorology | MPI-M |
| MRI-ESM2-0 | Meteorological Research Institute | MRI |
| NESM3 | Nanjing University of Information Science and Technology | NUIST |
| NorCPM1 | NorESM Climate modeling Consortium | NCC |
| NorESM2-LM | NorESM Climate modeling Consortium | NCC |
| NorESM2-MM | NorESM Climate modeling Consortium | NCC |
| SAM0-UNICON | Seoul National University | SNU |
| TaiESM1 | Research Center for Environmental Changes, Academia Sinica | AS-RCEC |
| UKESM1-0-LL | Met Office Hadley Centre | MOHC |

Table 1: List of CMIP models and their respective modelling institutions.

## 3 Analysis

In this study, we use Empirical Orthogonal Function analysis to identify the three primary modes of variability in sea level pressure in the ERA-20C and NOAA-20CR reanalyses. The first of these modes is considered to represent the North Atlantic Oscillation. We focus on the North Atlantic region, defined here as -90E to 40E longitude and 20N to 80N latitude (Figure 1 inserts). The analysis is applied to a 30-year moving window over the full time period of 1900 to 2010, with resulting times series plotted for the middle of each 30-year window, i.e. from 1915 to 1995. We use the 'rule of thumb' proposed by North et al. (1982) to determine if a particular mode is distinct from neighbouring modes based on the separation of their eigenvalues compared to the standard error for their eigenvalues. Where the standard error overlaps the eigenvalue of another mode, both are considered to be blended modes and not independent.

When attempting to compare EOF patterns from observations with those in models, two problems occur. Firstly, the modes may occur in a different order or may be different modes entirely, for example, the first mode in a particular model may not





represent the NAO. Second, the modes may be inverted compared to the observations. While the second issue is easily resolved
by identifying and inverting modes as needed, in this study we use an alternative approach to compared the modes called the
Common Basis Function (CBF; Lee et al., 2019), which circumvents both of the above issues. The CBF approach projects
model anomalies onto the EOFs of the reanalysis. This approach more directly addresses the question of how well is the
variability of a particular mode in the reanalysis represented in the model. A more detailed explanation of the CBF approach is

given in Lee et al. (2019) along with sensitivity testing of the method compared to the conventional EOF approach. The steps
to apply the CBF approach are as follows:

1. Calculate EOF from reanalysis and normalize to unit variance

2. Calculate the dot product of the spatial pattern of anomalies in the models and EOF pattern from reanalysis to get the
   unnormalized CBF PC time series

3. Calculate linear regression between CBF PC and temporal anomalies at each grid point to obtain the slope of regression
   at each grid point

4. Multiply regression slopes by the value of CBF PC at each point to maximise the variance associated with the simulated
   expression of reanalysis pattern

5. Calculate area-weighted mean of temporal variance at each grid point to derive total variance explained by a given mode

6. Multiply regression slopes by standard deviation of CBF PC to obtain the pattern of anomalies

## 4  Results

The EOF analysis of sea level pressure in the ERA-20C and NOAA-20CR reanalyses using a 30-year moving window shows
that the first three modes are very similar between the two reanalyses (Figure 1). This emphasizes the robustness of the findings
of how these modes have changed given that these two reanalysis products use different underlying models, spatial resolutions,

data assimilation methods, and there are also differences in the observations assimilated. Despite this they both capture the same
change in the relative importance of the first three modes of variability in this region. In both reanalyses, the first mode rises
from explaining approximately 32% of the variability in the early 20[th] century to 53% by the end of the century. Simultaneously,
the second and third modes explain a decreasing percentage of the variability in North Atlantic sea level pressure, dropping
from approximately 28% to 16% and from 14% to 10% for the second and third modes respectively. This shows a transference

of variability explained from the North Atlantic ridge and Scandinavian Blocking patterns into the NAO over the course of the
20[th] century. The bulk of the increase in percentage variability explained by the NAO occurs from the late 1950s onwards. This
increase suggests that the NAO index has become a more valuable tool for explaining and/or predicting changes in climate over
the North Atlantic and downstream over Europe in recent decades since the NAO has increased in the amount of variability in
sea level pressure explained by the pattern.







**Figure 1.** Percentage of variance explained by the first, second and third modes in both ERA20C (blue) and NOAA20C (orange). The modes are shown from dark to light shades for modes 1 to 3 respectively. The inserts show the pattern in sea level pressure associated with the first mode of variability in ERA20C for the 30-year periods centered on 1920 (left) and 1930 (right).

It is clear from Figure 1 that the first and second modes explained approximately the same amount of the variability in North Atlantic sea level pressure around 1920. When we examine the uncertainties using the North et al. (1982) 'rule of thumb', we find that the first mode is not distinct from the second mode until around 1930. The two inserts show the pattern associated with the first mode for the years 1920 and 1930. The 1930 pattern is the classical NAO pattern with a dipole of two centres of action, although they are spread horizontally due to the variation in the location of the centres of action over such a long time

period. The pattern for 1920 shows a single centre of action to the west of the U.K. This pattern does not match with the NAO pattern, nor with the North Atlantic Ridge pattern, but is likely a merging of the two as the EOF analysis has not separated the modes correctly.



To explore how well this shift in the variability explained by the NAO is reproduced by the current generation of climate models, we use CBF analysis on a 30-year moving window for each of the 50 CMIP6 models listed in Table 1. The CBF
analysis uses the first principle component from the ERA-20C EOF analysis. Since PC1 is not clearly separated from PC2 until after 1930, we limit our analysis of the CMIP6 models to 1930 onward. The models do not in general reproduce the variation over time seen in the reanalyses, but this is to be expected since there is no matching of the phase of internal variability between the simulations or between the simulations and the reanalyses (Figure 2). For example, the two versions of the Norwegian Earth System model (NorESM2) are the same model but run with a high or low resolution for the atmosphere and land components
(~1°and ~2°respectively), yet they show very different trajectories over the 20th century. If the increase in percentage variability explained by the NAO seen in the reanalyses were the result of external forcing, such as a warming climate, the lack of consistent trend or variation in the models would indicate that they fail to capture this response. This is highlighted by the multi-model mean which is approximately constant, or even slightly decreasing, over the 20th century. The lack of any consistent trend in the models supports the idea that the change in percentage variability explained is due to natural variability. In general, the
climate models over-estimated the importance of the NAO in the first half of the century and under-estimated its importance in the second half, when compared to the reanalyses. While the time series shown in Figure 2 were produced with CBF analysis, a similar picture is obtained when examining the first mode of variability using EOF analysis on each of the models independently (See Supplemental Figure S1).

If we accept that the increase in percentage variability explained seen in the reanalyses is due to natural variability, it is
interesting to determine how well the models reproduce the range of explained variability compared to reanalysis. The climate models show a large range in the percentage variability explained, from as low as 17% in ACCESS-ESM1-5 to as high as 67% in NorESM2-MM (Figure 2). Of particular note are the NorESM2-LM and AWI-ESM-1-1-LR models that never drop below 44% of the variability being explained by the NAO. The spread, as given by the interquartile range, in percentage variability explained in the ERA-20C reanalysis and individual models is shown in Figure 3. 46 of the 50 models underestimate the
spread when compared to reanalysis. NorESM2-MM is one of the few examples of a model having a greater spread than the reanalysis. This shows that the importance of the NAO pattern for explaining variations in North Atlantic sea level pressure is too consistent in the models and does not vary as much as in the reanalyses.

It is unclear from Figure 2 if any individual model is consistently producing a NAO similar to that found in the reanalyses. To explore this, we compared the percentage variability explained, pattern correlation, and Taylor skill score between different
30-year periods. The results showed that there was no consistency between the different periods. For example, Figure 4 shows the percentage variability explained between the initial and final 30-year periods from Figure 2. The models are ranked by the percentage variability explained during the initial period, and there is no indication that this ordering is preserved in the later period. This highlights that large-scale patterns in models should not be compared to reanalyses or observations for individual periods, since a model that compares well to the observations during one period cannot be assumed to compare well during
another period, and any good comparison is the result of coincidental phasing of internal variability. Similar plots for the first and last 30-year period are given for the percentage variability explained, pattern correlation, and Taylor skill score in Supplemental Figure S2, S3, and S4 respectively).



**Figure 2.** Percentage of variance explained by the first mode of EOF in the reanalyses and CBF in the CMIP6 models in a moving 30-year period. ERA20C and NOAA20c reanalyses are shown in bold in blue and orange respectively. CMIP6 models are shown by thin lines in grey, except for NorESM2-LM (red) and NorESM2-MM (blue). Multi-model ensemble mean is shown in thick black line.

Given the increase in percentage of variability in sea level pressure explained by the NAO found in the reanalyses, it is of interest to see how the variability in sea level pressure itself has changed over the same period. Figure 5 shows the area-weighted

mean variance in sea level pressure over the North Atlantic region for a 30-year moving window in both the two reanalyses and the 50 CMIP6 models. The reanalyses show a general increase, especially after 1950, rising from approximately 800 hPa at the beginning of the century to approximately 1000 hPa by the end of the century. This is equivalent to the standard deviation in sea level pressure changing from approximately 28 hPa to 31 hPa. This indicates a small but steady increase in either the depth or frequency of low pressure systems moving along the North Atlantic storm tracks (Feser et al., 2014).




**Figure 3.** Spread in the percentage variability explained by the first mode of EOF in the reanalyses and CBF in the CMIP6 models in a moving 30-year period, ranked in order of the interquartile range. Outlier points beyond 1.5 x interquartile range beyond the upper and lower quartiles are not shown. ERA-20C is shown in black, and the CMIP models are shown in grey except for NorESM2-LM (red) and NorESM2-MM (blue).

The climate models show quite a different picture, with a large spread in the variability of sea level pressure, ranging from 475 hPa in INM-CM4-8 to 1,633 hPa in IPSL-CM6A-LR. This suggests that some models have too few or weaker low pressure systems, while others have too many or deeper low pressure systems compared to the reanalyses. It is interesting to note the similarities between the changes of sea level pressure variance and variance explained by NAO in both the models and reanalyses (c.f. Figure 5 and Figure 2). For example, NorESM2-MM shows high values in both plots until the 1940s, when it

drops to more moderate values and remains approximately constant until the end of the century. This means that the change in the sea level pressure variance associated with the NAO pattern over this period is even larger than would be assumed from Figure 2.



**Figure 4.** Percentage of variability explained by the first mode from the Common Basis Function for each of the models in 30-year periods at the start and end of the full time series, 1915-1944 (top) and 1981-2010 (bottom). Models are ordered in both plots by the amount of variability explained for the initial period of 1915-1944. The horizontal black line in both plots gives the amount of variability explained by the first mode of EOF in ERA 20[th] Century reanalysis, upon which the CBFs are based. CMIP6 models are shown in grey, except for NorESM2-LM (red) and NorESM2-MM (blue).

## 5 Conclusions

In this study we have examined how the percentage of variability in sea level pressure explained by the North Atlantic Oscilla-
tion pattern has increased over the 20[th] century, while the percentage variability explained by the second and third modes has decreased. The latest generation of climate models do not reproduce this change, suggesting that the observed change is not the result of external forcing such as a warming of the climate, but is due to internal variability. However, most of the models also underestimate the spread in explained variance when compared to the reanalyses, indicating that the relative strength of



**Figure 5.** Area-weighted mean variance in sea level pressure in moving 30-year period. ERA20C and NOAA20c reanalyses are shown in bold in blue and orange respectively. CMIP6 models are shown by thin lines in grey, except for NorESM2-LM (red) and NorESM2-MM (blue). Multi-model ensemble mean is shown in thick black line.

the NAO is too persistent in the models. Examination of the variance in sea level pressure shows that the models have a greater

spread than is seen in the reanalyses, suggesting that the low pressure systems are too few or too weak in some models, and too strong or too deep in others. If the changes to the relative importance of the NAO seen in the reanalysis are indeed due purely to natural variability then this has direct implications for the ability of climate models to capture natural variability in climate extremes over Europe. It is also crucial that multidecadal prediction of changes to climate extremes over Europe account for the phase of natural variability in the relative strength of the NAO.

Another possibility is that the changes seen in the reanalysis are a combination of natural variability and a forced response that is not represented in CMIP6 models. This might explain why the models are systematically biased towards underestimating



the variability in the explained variance of the NAO. If part of these changes in the relative strength of the NAO are indeed due to a forced response that is lacking in climate models then there is a risk that there is a systematic underestimation of the changing risks of climate extremes over Europe in a warming world.

While the large-scale patterns derived using EOF analysis are a good indicator of the changes in weather systems over the North Atlantic, recent works have found that jet regimes are better at capturing spatial structure compared to patterns like the NAO, and have the advantage of a greater physical connection to the underlying weather systems (Madonna et al., 2021). Future work is planned to investigate how these jet regimes have varied over time in the reanalyses and how well the CMIP6 models capture these variations.

*Author contributions.*  Outten: Analysis and paper writing. Davy: Paper writing.

*Competing interests.*  The authors confirm there are no competing interests.

*Acknowledgements.*  This publication was supported by the KEYCLIM project, funded through Grant Number 295046. We acknowledge the World Climate Research Programme, which, through its Working Group on Coupled Modelling, coordinated and promoted CMIP6. We thank the climate modeling groups for producing and making available their model output, the Earth System Grid Federation (ESGF) for archiving

the data and providing access, and the multiple funding agencies who support CMIP6 and ESGF. Support for the NOAA Twentieth Century Reanalysis Project dataset is provided by the U.S. Department of Energy, Office of Science Innovative and Novel Computational Impact on Theory and Experiment (DOE INCITE) program, and Office of Biological and Environmental Research (BER), and by the National Oceanic and Atmospheric Administration Climate Program Office.



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
