# Peer review of "Changes in the North Atlantic Oscillation over the 20th Century"

_EGUsphere, 2023_

## Author Comment (AC1)

**Response to Reviewer 1**

We'd like to thank the reviewer for the comments, and particularly for their time and effort in undertaking this review. We have responded to the individual points below.

General comments:

I understand that the authors focus on the winter NAO only, but it is not clearly stated in the text. Could the authors clearly specify that they focus on the winter season only in the introduction and specify which months are downloaded in section 2. If I am wrong and all the seasons are considered, is this change in explained variance due to one season in particular or is it observed in every season?

This is a good point and well made, we should have specified that we are focussing on the wintertime NAO. We'll add a sentence in the introduction making it clear that "Since the NAO dominates the climate variability over the North Atlantic-European sector during the wintertime, we focus our study on the winter months, defined here as December-January-February (DJF)."

Regarding the other seasons, it has not been thoroughly investigated how the variance explained varies in other seasons. We did examine the summertime (JJA), which shows a similar but weaker trend in the NAO variance explained, with an accompanying weaker decrease in the EOF2. No tests were made for the significance of that trend however, so it may be negligible. By eye, the variance explained rises from around 28% to around 33% for EOF1, with EOF2 dropping from approximately 21% to approximately 16%.

Why the authors did not use more recent reanalyses? Like ERA5 instead of ERA-20C, that spans 83 entire years (1940-2022). And can we fully trust ERA-20C/NOAA-20CR in 1930?

We were originally interested in the early 20th century as part of the study, hence the use of the 20th Century reanalyses. We have also examined these results in ERA5, which gave almost identical results to those seen with ERA-20C back as far as 1940.

It is a very valid concern about the quality of the 20th Century reanalyses before 1930. We believe the limitations of reanalyses before 1950 are widely known, and we left it to the readers' good judgement to evaluate the reliability of the results in the early 20th century. However, we could add a sentence highlighting this concern if it is felt necessary.

The authors show that the observed increase in explained variance is not visible in the CMIP6 ensemble mean, which could mean that it is not a consequence of an external forcing like global warming. Have the authors thought of any other process that could explained this change in explained variance on such a long timescale? Could the authors comment more on this in the conclusion?

We have not investigated in any great detail or to any conclusion what other process could explain the CMIP6 ensemble failing to reproduce the observed trend. Looking at the ensemble of models, the range of variability explained is comparable to the observed change in variance explained. All models are provided with the same forcing for the 20th century and not just for GHGs, and yet the models show a spread in response including increasing,

decreasing, no change, increasing then decreasing, etc. Given these two facts together, we assert that the change in variability is not a response to the applied forcing and therefore assume it is the result of internal variability, as noted in the paper.

Speaking for myself, I would speculate that it is possible that this could be the results of a long term, multi-decadal variability (perhaps with a variability of 60-80 years such as that found in studies on Bjerknes compensation). If the models reproduced such oscillations and were initialised in different phases of the multi-decadal signal, this could possibly produce the observed differences in the long term changes in variance explained by the NAO. However, it would be very difficult to reliably confirm a 60-80 oscillation in a 150 year simulation since you would barely have two full cycles. This is speculation and would need significant investigation to confirm.

Specific comments:

Figure 1: I understand that the shading represents the standard error, but it does not appear in the caption. Please add an explanation for the shading.

This was an oversight and we will update the caption to include an explanation of the shading. Thank you for pointing this out.

Figure 3: I do not understand why the spread in explained variability reaches negative values. Are the boxplots "shifted" so that the median is zero for each model and ERA-20C? Could the authors better describe in the caption what is plotted in Figure 3?

The box plots are shifted so that the medians are zero for each model and for ERA-20C. We will add a better description in the caption.

Plots in the Supplementary Materials: why do the authors highlight with green a third model?

The two models highlighted are the NorESM2 model run with medium (MM) and low (LM) resolutions. The green line in the supplemental figure is for NorCPM, another version of NorESM (specifically NorESM1 but with many of the upgrades made between versions 1 and 2). NorCPM is normally run with an assimilation step and is a Climate Prediction Model. We will update the bar charts to remove the green highlighting.

---

## Author Comment (AC2)

**Response to Reviewer 2**

We appreciate the reviewer's time and effort in undertaking this review, and would like to thank them for their comments. We have responded to the individual points below.

**General comments**

This paper examines CMIP6 models' reproducibility on interannual variability mode, NAO. The study has found increasing trend of the percentage of explained variability by the NAO in the reanalysis datasets, while that is not the case in many of CMIP6 models.

I believe the scope of the research meets the scope of the journal, and the paper was thoughtfully prepared. One concern I have though is about the percentage of the explained variability (PEV) The authors mostly focus on the PEV by the NAO. While I acknowledge that it is an important measure, it could be sensitive to the magnitude of the total variance of the model because the PEV is relative value to the total variance. I think having this addressed in addition would make the paper more robust.

We agree completely with the reviewer that percentage variability alone could be misleading. It is for this reason that we included in the paper the change in total variance in sea level pressure over the same 30-year moving window. As you highlight, it is a portion of this total variance that is explained by the percentage variability explained. These two fields together provide the total variance explained. The three plots of percentage variance explained, total variance explained, and variation in the sea level pressure are all quite similar. We didn't feel it was necessary to show all three. Obviously we wanted to show the percentage variance explained as this is a focus in the paper, however, the question then arose of which of the other two to show in the paper, and we opted for the field itself, being the total variance in sea level pressure.

To address your very reasonable concern, we have now added the plot of total variance explained into the supplemental material and made reference to it in the text.

In the discussion section (line 180), it was discussed that the trend from the reanalysis datasets could be resulted by the combination of the natural variability and the forced response. I wonder if this point could be explored more. One possible path could be working with anomaly monthly PSL field by removing any linear trend in it. This might be helpful to provide some insights on isolating influence of forced response.

Thank you for the suggestion. We do propose in the paper that since the models all have a similar forcing but do not show any consistent response, the trend in the reanalysis is likely the result of internal variability. However, this is not the only possibility and at the request of your fellow reviewer, we have included some discussion of other possibilities. We are currently undertaking some analysis for a follow up paper, however, that focusses primarily on the changes in NAO and its links to precipitation as seen in the climate models. We may at some point pursue this matter further and attempt to demonstrate more robustly that this change is the results of internal variability, and your proposed approach could certainly help with that. At this time however, we are happy for interested readers to pursue the investigation for themselves.

Specific comments

Line 74 "times series": Does this be replaced by "time series"?

Thank you for spotting this typo, we will correct it.

Fig 1: What are the shadings indicating?

This was an oversight on our part, we will update the captions on all the figures to provide a more complete explanation of what is shown.

Table 1 and analysis: Is one ensemble member of each model used for the paper? Please consider clarify this in the paper.

We can add a statement to clarify this point.

Line 130 "In general, the climate models over-estimated the importance of the NAO in the first half of the century and under-estimated its importance in the second half, when compared to the reanalyses.":

I am not sure what "importance" indicates in this description. Does it indicate the "portion of variability explained by the NAO"?

As you suggest, we are referring to the portion of variability explained by the NAO. We will revise to make this more clear.

Also, this is the percentage, not the magnitude of variance itself. I think the magnitude of variance for each 30-year epoch might also need to be analyzed to complement the analysis for the "percentage of variability explained by NAO"

This refers back to your more general point regarding percentage versus absolute magnitude of variance explained. Given the similarity of the three plots of total variance in sea level pressure, percentage variance explained, and total variance explained, we have now included the total variance explained in the supplemental material and referenced it within the paper text, as discussed above.